# Phage Therapy in a Burn Patient Colonized with Extensively Drug-Resistant *Pseudomonas aeruginosa* Responsible for Relapsing Ventilator-Associated Pneumonia and Bacteriemia

**DOI:** 10.3390/v16071080

**Published:** 2024-07-05

**Authors:** Cécile Teney, Jean-Charles Poupelin, Thomas Briot, Myrtille Le Bouar, Cindy Fevre, Sophie Brosset, Olivier Martin, Florent Valour, Tiphaine Roussel-Gaillard, Gilles Leboucher, Florence Ader, Anne-Claire Lukaszewicz, Tristan Ferry

**Affiliations:** 1Centre des Grands Brûlés Pierre Colson, Hôpital Edouard Herriot; Lyon, Hospices Civils de Lyon, 69003 Lyon, France; jean-charles.poupelin@chu-lyon.fr (J.-C.P.); olivier.martin@chu-lyon.fr (O.M.); anne-claire.lukaszewicz@chu-lyon.fr (A.-C.L.); 2Pharmacie de Centre Hospitalier Nord, Hôpital de la Croix-Rousse, Hospices Civils de Lyon, 69002 Lyon, France; thomas.briot@chu-lyon.fr (T.B.); gilles.leboucher@chu-lyon.fr (G.L.); 3Service de Maladies Infectieuses et Tropicales, Hôpital de la Croix-Rousse, Hospices Civils de Lyon, 69002 Lyon, France; myrtille.le-bouar@chu-lyon.fr (M.L.B.); florent.valour@chu-lyon.fr (F.V.); florence.ader@chu-lyon.fr (F.A.); 4Phaxiam Therapeutics, 60 Avenue Rockefeller, Bâtiment Bioserra, 69008 Lyon, France; cindy.fevre@phaxiam.com; 5Service de Chirurgie Plastique et Reconstructrice, Hôpital Edouard Herriot; Lyon, Hospices Civils de Lyon, 69003 Lyon, France; sophie.brosset@chu-lyon.fr; 6Faculty of Medicine, Université Claude Bernard Lyon 1, 69100 Villeurbanne, France; 7Centre International d’Infectiologie, Inserm U1111, CNRS UMR5308, ENS de Lyon, UCBL1, 46 Allée d’Italie, 69007 Lyon, France; 8Institut des Agents Infectieux, Hôpital de la Croix-Rousse, Hospices Civils de Lyon, 69002 Lyon, France; tiphaine.roussel-gaillard@chu-lyon.fr; 9Education and Clinical Officer of the ESCMID Study Group for Non-Traditional Antibacterial Therapy (ESGNTA), 4051 Basel, Switzerland

**Keywords:** *Pseudomonas aeruginosa*, antimicrobial resistance, phage therapy, burns, ICU, ventilator-associated pneumonia

## Abstract

*Pseudomonas aeruginosa* is one of the main causes of healthcare-associated infection in Europe that increases patient morbidity and mortality. Multi-resistant pathogens are a major public health issue in burn centers. Mortality increases when the initial antibiotic treatment is inappropriate, especially if the patient is infected with *P. aeruginosa* strains that are resistant to many antibiotics. Phage therapy is an emerging option to treat severe *P. aeruginosa* infections. It involves using natural viruses called bacteriophages, which have the ability to infect, replicate, and, theoretically, destroy the *P. aeruginosa* population in an infected patient. We report here the case of a severely burned patient who experienced relapsing ventilator-associated pneumonia associated with skin graft infection and bacteremia due to extensively drug-resistant *P. aeruginosa*. The patient was successfully treated with personalized nebulized and intravenous phage therapy in combination with immunostimulation (interferon-γ) and last-resort antimicrobial therapy (imipenem-relebactam).

## 1. Introduction

*Pseudomonas aeruginosa* is a Gram-negative pathogen widely prevalent in hospital settings. It is one of the main causes of healthcare-associated infection in Europe, mostly in intensive care units (ICUs) following ventilator-associated pneumonia (VAP) and/or bacteremia [1]. The common risk factors for *P. aeruginosa* bacteremia are in-hospital patients, invasive ventilation, intravenous or intra-arterial catheters, and surgical wounds or severe burns [2]. Skin or respiratory colonization is also a risk factor for developing an infection [3]. *P. aeruginosa* represents a significant burden in burn ICUs, especially in patients with extensive burns (>20–25% of the total body surface area), for the following reasons: (i) these patients are susceptible to be colonized by environmental bacteria such as *P. aeruginosa*; (ii) they are at risk of skin graft infection and necrosis; (iii) they are at risk of VAP due to their frequent exposure to long-term mechanical ventilation; and (iv) they suffer from immune deficiency [4,5]. Of note, 19% of patients admitted to a French burn center in 2010 suffered from at least one episode of infection, with this rate increasing to 48% if the burn surface area exceeded 30% [6]. In their 2017 study of healthcare-associated infections in burn patients, Strassle et al. found the 60-day infection incidence to be 8% (with a higher risk associated with flame and electrical burns) [7]. 

A recognizable but not specific characteristic of wound colonization or infection by *P. aeruginosa* is the yellow/green color of the infection site and a malodorous fruity smell [8]. 

Invasive infection is now the main cause of morbidity after burn injuries and is responsible for half of deaths in these patients [9]. The European prevalence of pneumonia in ventilated patients (not only burn patients) is 19.4% [3,10]. VAP is reported in 5 to 40% of ventilated patients after 2 days of mechanical ventilation, with a peak between days 5 and 9 [11]. The risk increases with the ventilation duration. *P. aeruginosa* can be isolated in 10 to 40% of VAP cases, while it can worsen mortality by up to 32 to 42.8% [1,12,13,14,15]. Mortality also increases when the initial antibiotic treatment is inappropriate, especially if the patient is infected with *P. aeruginosa* strains that are resistant to many antibiotics [12]. In the ICU, 18.6% of these strains are found to be resistant to three or more antimicrobials, leading to their classification as multidrug-resistant (MDR) [1]. Source control is an important factor for clinical success and is predictive of in-hospital mortality [16,17]. 

*P. aeruginosa* is part of the so-called ESKAPE pathogens, which includes *Enterococcus faecium*, *Staphylococcus aureus*, *Klebsiella pneumoniae*, *Acinetobacter baumannii*, *Pseudomonas aeruginosa*, and *Enterobacter species*. [18]. Antimicrobial resistance is a slow-motion tsunami, mostly if those pathogens acquire an antimicrobial resistance gene, and infections due to MDR *P. aeruginosa* represent a significant burden, especially because resistant strains spread rapidly in the hospital setting due to the reduction in treatment options [12]. The European Center for Disease Prevention and Control, which publishes an annual epidemiological report on antibiotic resistance in Europe, found that in 2021, 31% of collected P. aeruginosa strains were resistant to at least one of the antimicrobial groups tested (piperacillin-tazobactam, fluoroquinolones, ceftazidime, aminoglycosides, and carbapenems) [1,19], with an increasing incidence of carbapenem-resistant strains over time. In the United States, between 2018 and 2020, around 15,4% of *P. aeruginosa* isolated in VAP in the ICU were MDR, as reported by the National Healthcare Safety Network [20]. In 2017, the World Health Organization classified carbapenem-resistant P. aeruginosa as one of the three critical pathogens for which antibiotics research and development should be promoted [21]. Carbapenem resistance has a significant negative impact on mortality during P. aeruginosa invasive infections. The average 30-day all-cause mortality rate after the index culture of patients infected with a carbapenem-resistant strain of P. aeruginosa was 22% vs. 12% with non-carbapenemase-producing strain infection in the Lancet published trial of Reyes [22]. 

Phage therapy is an emerging option to treat severe *P. aeruginosa* infections. In general, it involves using natural viruses called bacteriophages (phages), which have the ability to infect, replicate, and, theoretically, destroy their host bacteria [23]. They are developed as lytic phages that target a specific strain of a *P. aeruginosa* population that infects the patient and destroy their host. Phages are ecologically safe because they replicate inside a specific host bacterium (as well as in a biofilm) and do not infect nonpathogenic bacterial species. Phage therapy was previously evaluated in a randomized clinical trial in burn patients with wounds infected by *P. aeruginosa*, while some case reports describe patients with other types of severe infections such as sepsis and bacteremia, prosthetic joint infections, and lung infections [24,25,26,27]. Among the case reports of phage therapy against lung infections caused by MDR P. aeruginosa, phages were administered by intravenous and inhalation routes. In all cases, antibiotics alone led to therapeutic failures and second- or third-line antibiotics were used in combination with phage therapy. No side effects were recorded, and in all cases, the clinical outcome was favorable and no relapses due to *P. aeruginosa* occurred. In some cases, *P. aeruginosa* became resistant to some phages but still resulted in overall clinical resolution [28]. 

We report here the case of a severely burned patient who experienced relapsing VAP associated with skin graft infection and bacteremia due to extensively drug-resistant (XDR) *P. aeruginosa*. The patient was successfully treated with personalized nebulized and intravenous phage therapy in combination with immunostimulation (interferon-γ) and last-resort antimicrobial therapy (imipenem-relebactam).

## 2. Case Presentation

A 52-year-old male was admitted to the burn center after suffering self-inflicted burns from setting fire to a gas can with a suicidal intention. The patient’s relevant medical history was hypothyroidism, tobacco-related chronic obstructive pulmonary disease treated with bronchodilatators (salbutamol and fluticasone-salmeterol), and severe depressive syndrome treated with venlafaxine and cyamamezine. The patient had a recent past history of psychiatric hospitalization after voluntary drug overdoses. 

On his arrival at our burn center, the patient required intensive vascular resuscitation and sedation, leading to orotracheal intubation. The depth and extent of the burn injuries covered 81% of the total body surface area with circumferential impairment of the four limbs. On admission, a rectal carriage with an XDR strain of *Enterobacter cloacae* with a phenotype of New Delhi metallo-β-lactamase (NDM-1)-producing strain was discovered. The patient underwent a total of 21 surgical interventions for burn excision and cutaneous grafting with autografts and allografts. During his stay, he experienced several successive episodes of septic shock, which required the use of vasoactive drugs such as noradrenaline due to the inefficiency of vascular filling, although this led to multiple complications. He also experienced acute kidney insufficiency (stage 3 according to the Kidney Disease: Improving Global Outcomes [KDIGO] classification), which required transient renal replacement therapy. The first etiology for kidney injury was pre-renal associated with initial hypovolemic shock, while the second phase that indicated renal replacement therapy was considered to be organic due to the nephrotoxic medication intravenous colistin [colistimethate sodium], administered at a dose of 3 MUI every 6 h on day 1 and then 3 MUI every 8 h for 7 days, combined with septic shock, which was due to VAP associated with bacteremia caused by carbapenem-resistant *P. aeuroginosa* (CRPA). After several other infectious events with an iterative VAP, the patient received various antibiotics: first, amoxicillin for *Haemophilius influenzae*, followed by piperacillin-tazobactam for a second episode of *Streptococcus pneumoniae* and then ceftazidime-avibactam for 7 days to treat *XDR E. cloacae* with New Delhi metallo-β-lactamase.

Due to the severity of septic shock and his severe immunosuppression measured by the monocyte expression of human leucocyte antigen–DR isotype (HLA-DR), the patient received a first immunostimulation with subcutaneous interferon-γ 100 mcg once a day for 7 days, but with no clinical or biological improvement. 

He rapidly developed a new VAP with classification based on the degree of hypoxemia described by the ratio of the partial arterial pressure of oxygen (Pa02) over the fraction of inspired oxygen (Fi02)—Pa02/Fi02 of 408—or a clinical pulmonary infection score which is specific for VAP (CPIS based on fever, leucocytosis, extent of oxygenation impairment) of 7 [29,30]. This was the first episode with polymicrobial flora and the first due to XDR *P. aeruginosa* (associated with the similar XDR *Enterobacter cloacae* and *Klebsiella pneumoniae*) (Figure 1).

The antibiogram of this first *P. aeruginosa* VAP is detailed in Table 1. The patient was treated intravenously with aztreonam and ceftazidime/avibactam (to obtain the aztreonam-avibactam combination, as avibactam was not available without ceftazidime) for 15 days. Unfortunately, the patient experienced a relapse with a new VAP that evolved into a severe clinical form of adult respiratory distress syndrome according to the Berlin definition [29,30] (Pa02/Fi02 of 86 and CPIS of 9) associated with skin colonization, graft lysis, and bacteriemia caused by a more resistant strain of *P. aeruginosa* (Table 1), which required complex antibiotic treatment with intravenous and inhaled colistin (3 MUI/8 h) and intravenous imipenem-cilastatin-relebactam (500 mg/6 h) for 15 days. Given the repeated septic shock in a severely burned patient, his immune profile was monitored weekly with lymphocyte typing and monocyte HLA-DR. This monitoring revealed severe and persistent immunosuppression that justified a second immunostimulation with interferon-γ. 

By then, the patient had already been hospitalized for 2 months and had undergone multiple skin excisions and grafts, which impacted his immunological status with proinflammatory stimulation. The poor evolution of these skin grafts was also considered to be due to cutaneous colonization with the same resistant strain of *P. aeruginosa*. Anticipating the recurrence of VAP and/or sepsis due to this *P. aeruginosa* strain, the isolate responsible for the second VAP was sent to Phaxiam therapeutics (previously named Pherecydes Pharma), a French company specialized in phage therapy, in order to test the *P. aeruginosa*-targeting phages PP1450, PP1777, PP1792, and PP1797 that are currently part of a research and development program. These phages have been the subject of previous publications [31,32,33,34,35]. They are natural virulent phages that were isolated from wastewater in 2014 and selected for their host range on different panels of strains, especially strains isolated from lung infections and MDR strains. The four phages belong to the *Caudoviricetes* class. The closest homologs of PP1450 and PP1777 in the Genbank database belong to the *Pbunavirus* genus, while those of PP1792 and PP1797 belong to the *Bruynoghevirus* genus based on the ICTV 2022. Their genomes (WO2015059298A1; WO2016071503A1) do not encode any virulence factor or antibiotic resistance gene. In the context of clinical development, the phages were produced under GMP grade and are supplied as “ready-to-use” vials by the company. Each vial contains 1 mL of a single phage type at a concentration of 1 × 10^10^ PFU/mL in a buffered solution. The manufacturing host was engineered by the company and do not encode any prophage nor toxin. The manufacturing process and analytical plan were submitted to the French National Agency for Medicines and Health Products Safety (*Agence nationale de sécurité du médicament et des produits de santé*—ANSM), and assessed in the scope of the compassionate use. The list of quality controls performed on the drug substances and drug products include Identity, Potency, Process, and Host impurities (including endotoxins), microbiological contaminants, and sterility. To evaluate the patient’s strains’ susceptibility to the phages, two methods were used. The first method is the plaque assay by spot test [33]. It is based on the visualization of a bacterial lysis when the strain is grown in a solid medium. In the case of bacterial lysis (measured in PFU), the efficiency of plating (EOP), defined by the ratio between the phage titer on the patient strain and the phage titer on its reference strain, was calculated, with the reference strain being a highly susceptible strain likely approaching the true phage concentration. When the score is close to 1, the phage is more efficient, indicating that it will be active at low concentrations. An EOP score of 0 indicates that no lysis range is visible. An EOP score >0.1 corresponds to a very efficient phage, an EOP score between 0.005 and 0.1 to an efficient phage, and an EOP score between 0 and 0.005 to a moderately effective phage [36]. The second method, known as “broth microdilution assay”, determines the minimum concentration of phages inhibiting 80 ± 8% of bacterial growth (MIC_80_). The strain is grown in a 96-well plate at a starting concentration of 5 × 10^5^ to 1 × 10^6^ colony-forming units (CFU)/mL in association with seven phage concentrations from 1 × 10^3^ PFU/mL to 1 × 10^9^ PFU/mL. Bacterial concentration is monitored over time by optical density at 600 nm until the phage-free condition reaches the stationary phase. With the plaque spot assay method, the PP1792 phage showed a partial lysis of the spots without PFU visualization, suggesting a mechanism of lysis without an abortive system. The minimal concentration of phages that induced this lysis was 7.67 Log PFU/mL. PP1450, PP1777, and PP1797 did not show any lytic activity on the strain in the plaque spot assay. With both microdilution assays, the MIC_80_ of PP1792 and PP1797 was 4.22 and 6.02 Log PFU/mL, respectively. These latter phages were considered active on the patient’s strain, whereas PP1450 and PP1777 were considered inactive. In the case of subsequent *P. aeruginosa* infection, the phages PP1792 and PP1797 were selected as candidates to treat the patient. As anticipated, after terminating the treatment of the second episode, the patient developed a third VAP (Pa02/Fi02 of 395, CPIS of 7) caused once again by this XDR strain of *P. aeruginosa* with bacteremia. For this episode, imipenem-cilastatin-relebactam (500 mg/6 h) with inhaled colistin (3 M UI/8 h) was initiated once again two days before, as the strain was still susceptible to these antibiotics (Table 1 and Figure 2). A new phagogram was performed and the results were similar; the strain presented the same susceptibility to the phages. Phage therapy was thus set up as adjuvant therapy due to the severe and recurrent nature of the infection, the immunocompromised status of the patient, and the use of last-resort antimicrobial therapy. The antibiotic combination was initiated for two days before the phage administration and continued for a total of 21 days. The strain of *P. aeruginosa’s* antibiogram was known after 48 h and matched the already developed phages that were then administered.

Under the supervision of the French National Agency for Medicines and Health Products Safety (*Agence nationale de sécurité du médicament et des produits de santé*), we discussed the indication for phage therapy, with the active phages PP1792 and PP1797 that were provided by Phaxiam to our hospital pharmacist at the Hospices Civils de Lyon. The patient and his family received all the necessary information about phage therapy before its use, and the patient’s family gave written consent to treat. Phage therapy consisted of a 7-day course of treatment with the following: (i) three inhaled doses (one dose every 2–3 days) of the personalized cocktail of phages targeting *P. aeruginosa*, which was diluted in 5 mL of NaCl 0.9% (each phage at 2 × 10^9^ PFU/mL) and administered using a vibrating mesh nebulizer connected to the ventilator; and (ii) a daily 3 to 5 min direct intravenous injection of the same phage cocktail, diluted in 50 mL of NaCl 0.9% (each phage at 2 × 10^8^ PFU/mL) (Figure 3), for 7 days. 

During this treatment course, no adverse event attributable to phage therapy was observed, and the patient rapidly improved with reduced bronchorrhea and less oxygen dependence. After 14 days of imipenem-cilastatin-relebactam and 21 days of inhaled colistin, antibiotics were stopped. Additionally, the patient developed fungemia caused by *Candida parapsilosis* due to cutaneous translocation, which was treated with fluconazole for 6 weeks. He also developed bacteremia due to methicillin-susceptible *Staphyloccocus aureus* and septic thrombophlebitis at the site of one of his central venous catheters, which was treated with daptomycin for 6 weeks. 

One month after the VAP treated with phage therapy, the patient experienced a recurrence of pneumonia due to the same strain of *P. aeruginosa*, which was found in endotracheal aspiration and blood culture but with a less severe clinical form and with slightly increased oxygen needs (Fi02 40%). Imaging revealed limited pneumonia (Figure 4), with no hemodynamic support being required. The patient once again developed cutaneous graft lysis with the yellow-green color of the wound dressings.

Imipenem-cilastatin-relebactam (500 mg/6 h) was initially empirically initiated and then switched to Cefiderocol (1 g/8 h) adapted with the resistance profile when available, associated with inhaled colistin (3 M UI/6 h), two days before a new course of phage therapy was administered, with the same modalities as previously performed (Figure 3). The antibiotics were continued for 14 days. No adverse event attributable to phage therapy, such as fever, was recorded. Liver and renal function were tested twice a week during the infectious episodes with no disturbance recorded. The delay between the two phage therapies was 43 days. The patient was extubated on the last day of phage therapy followed by intermittent noninvasive ventilation for 13 days (BiPAP). Just before and during this treatment course, a significant increase in the monocyte expression of HLA-DR (mHLD-DR) was recorded (Figure 1). 

The final outcome was favorable. The patient was discharged from the ICU after 203 days of stay, including 155 days under mechanical ventilation. 

## 3. Discussion

The prevalence of bacterial infections in burn patients was around 19% of patients hospitalized in French burn treatment centers in the summer of 2006 [8]. Burns disrupt the protective cutaneous barrier, meaning that patients are constantly exposed to external pathogens. The lesion may thus be colonized by the patient’s cutaneous flora or by bacteria found in the ICU environment [5,37]. 

In burn patients, four types of pathogens are particularly feared: Clostiridium tetani in the initial admission phase, *Staphylococcus aureus* in the early phase or due to intravenous materials (central lines, arterial catheters), *Pseudomonas aeruginosa* in the secondary phase, and anaerobic bacteria that translocate from the digestive flora. In the pathophysiology of infection in burn patients, the low vascularization of the burned area due to ischemia and the occurrence of cutaneous necrosis or thrombosis increase the risk of skin infection, and due to the immunocompromised state of burn patients, there is a high risk of systemic bacterial dissemination [6,9]. Bacterial proliferation in the wound is mostly the first phase of local infection that leads to systemic dissemination [9,38]. Wound infection is considered if more than 10^5^ colony-forming units develop on the wound’s surface; this can evolve to a threatening situation for the surrounding healthy tissue and deteriorate to sepsis [38]. The definition of sepsis in the ICU should be adjusted for burn patients. They already present systemic inflammatory response syndrome (with tachypnea, hyperleukocytosis, tachycardia, hyperthermia, and systemic arterial hypotension) due to their wounds, which expose them to a high level of inflammatory mediators [38]. The Third International Consensus Definitions for Sepsis and Septic Shock of 2016 states that “Sepsis should be defined as a life-threatening organ dysfunction caused by a dysregulated host response to infection” [39]. Isolated leucocyte count is a poor marker of sepsis in burn patients due to their continuous inflammation, although it should be included in a more general clinical picture (neurological deterioration, increased oxygen requirement, increased vasopressor support, delayed wound healing, etc.) [40]. 

The initial systemic hyperinflammatory response is due to the extent of the burn injury itself, especially in the case of severe burns affecting more than 25% of the body surface area, followed by a prolonged phase of immunosuppression. Acquired immunodepression syndrome is described as a severe host immune response and is correlated with a high severity score on admission to a general ICU (based on SOFA score, APACHE II). This inflammatory response is essential for burn healing, although it also immediately stimulates the patient’s innate immune system for a prolongated period [41]. The patient’s immune and inflammatory response begins within 30 min after the injury and can continue for up to 2 years after the initial injury [42,43]. This inflammatory cytokine storm and the inability to restore a functional immune and inflammatory profile reduces resistance to severe infections and leads to poor outcomes [41,44,45]. Each surgical intervention destabilizes the pro-/anti-inflammatory balance by increasing the inflammatory response to graft excision and colonized cutaneous manipulation, which can lead to bacterial migration into the blood stream and thus infection. Cellular immunity is altered mostly because of the decrease in tissue perfusion caused by hypovolemia, while microthrombi, neutrophil chemotaxis, and lytic activity are compromised by the lack of oxygen delivery to the wound [45]. This down-regulation more specifically involves a decrease in the activity of natural-killer lymphocytes (reduced interferon-γ secretion), down-regulation of the complement system, a lack of activation of CD4+ lymphocytes, and a reduction in phagocytose due to decreased mHLA-DR [46,47]. This fall in mHLA-DR results in the insufficient antigen-presenting function of monocytes and impairs their capacity to induce the T-cell response, which forms part of the immediate immune response to aggression and is the precursor of macrophages [48]. The low mHLA-DR level (<30% or <8000 monoclonal antibodies bound per cell) is an interesting and standardized biomarker of immunosuppression, as it is associated with a higher risk of bacterial infections, septic shock, and mortality [42,44,49]. By targeting patients with a higher infectious risk, immunostimulant agents (interferon-γ) can be administered during the immunodepressive phase indicated by the low expression of mHLA-DR to enhance the patient’s immune system and thus reduce morbidity and mortality [46,47]. 

Burn infections caused by *P. aeruginosa* deteriorate rapidly, leading to systemic spread and high mortality within weeks. Sepsis is thus the leading cause of death in patients with severe burns [50]. *P. aeruginosa* has various pathways of resistance, including the creation of biofilm in which it can survive in extreme situations. This bacterium grows in a humid environment, meaning that a strict control plan for bacteria in the water of healthcare facilities with regular dosing is essential to allow for the screening and disinfection of environmental reservoirs at the origin of gene transmission [51]. *P. aeruginosa* is a pathogen responsible for late infections, which could be partially prevented by adequate antiseptic measures. Moreover, *P. aeruginosa* makes the infection’s treatment challenging on account of its rapid mutation and antibiotic resistance [52]. 

One of the specificities of cutaneous lesions due to *P. aeruginosa* is the alkaline pH of the junction zone of healthy/burned tissue (pH 9), which, unlike *S. aureus,* is propitious to the growth of *P. aeruginosa* due to the exudate of the wounds and the decreased concentration of iron. To overcome this limitation regarding the quantity of iron, *P. aeruginosa* overproduces pyoverdine to facilitate the acquisition of iron, which is essential for its development, thus giving the recognizable green color to wound dressings. However, it has been shown that pyocyanin also blocks wound healing by promoting conditions of low-level oxidative stress and inhibiting the MAPK pathway, which plays a crucial role in tissue repair [53]. *P. aeruginosa* damages tissue through the action of proteases such as elastase, which degrades elastin. Its production increases greatly in the exudate and is a sign of bacterial virulence [54]. 

Critical illness requiring prolonged intensive care and prolonged mechanical ventilation increases the risk of bacterial skin or lung colonization (by the biofilm formation of *P. aeruginosa* on respiratory epithelial cells), which can lead to infection with resistant strains [1]. Antimicrobial resistance represents a significant clinical challenge in patients infected with *P. aeruginosa*, as a susceptible strain may easily acquire antimicrobial resistance to various antibiotics. Indeed, *P. aeruginosa* is becoming increasingly resistant and difficult to treat, with the increase in mainly carbapenem-resistant *P. aeruginosa* and XDR and PDR strains [55,56]. Like other Gram-negative bacteria, the membrane of *P. aeruginosa* has a porin that channels, among others, certain antibiotics like β-lactams. In response to environmental stimuli such as the presence of antibiotics, *P. aeruginosa* can easily adapt porin expression and become resistant to almost all β-lactam available on the market by cumulating other mechanisms of resistance such as inducible AmpC cephalosporinase expression. The most well-known mutation is OprD in response to carbapenems, which leads to decreased sensibility or resistance to this antibiotic class. Another resistance mode involves altering the polarity of the outer cellular membrane, which contributes to decreased sensibility to aminoglycosides and polymyxins [57]. On the outer cellular membrane, *P. aeruginosa* can increase the expression of efflux pumps, which can reduce the susceptibility to many antibiotics such as fluoroquinolone and β-lactams without leading to strong resistance as an isolated mechanism [56]. To treat MDR *P. aeruginosa*, it is recommended to combine conventional antipseudomonal β-lactams (i.e., meropenem, imipenem, piperacillin-tazobactam, cefepime, and ceftazidime) with a second agent such as an aminoglycoside, fluoroquinolone, or polymyxin [58,59]. New alternatives, mostly a new molecule combined with β-lactam inhibitors, are now available and approved for activity against MDR and XDR *P. aeruginosa* (e.g., ceftolozane-tazobactam, ceftazidime-avibactam, imipenem-relebactam, and cefiderocol), although their combination with another active drug is often not feasible [54,56]. Colistin (colistimethate sodium) remains frequently active in high doses (6–9 million units/day), although the kidney toxicity of this drug limits its systemic use, which explains why inhaled colistin is frequently prescribed for patients with MDR or XDR *P. aeruginosa* VAP as salvage therapy [60]. Colistin is a cationic lipopeptide with a bactericidal action that interacts with the phospholipids of bacterial cell membranes and increases cell membrane permeability. Its narrow therapeutic window is caused by poor tissue penetration and can be highly toxic when systemically administered [61]. By nebulization, high pulmonary drug concentrations can rapidly be obtained locally with low systemic passage, which is useful in sepsis of pulmonary origin, although its use is still controversial [62,63,64]. It is mostly reported in patients with cystic fibrosis, although its increasing use in VAP-related MDR *P. aeruginosa* and *A. baumannii* has proven its effectiveness in ICU settings [63,64]. It has been reported that the effective sputum concentration is maintained longer after inhalation (up to 12 h) [62,63]. Dosage in aerosol devices needs to be higher to reach therapeutic concentration, but because of the reduced systemic exposure, it has a lower risk of systemic toxicity. The administration route can lead to adverse effects ranging from chest tightness to bronchospasm, mostly caused by osmolality and excipient content [61,63,65]. In the ICU, ventilator settings as well as the position and type of the aerosol generator also impact drug delivery [66,67]. No strong evidence is currently available in support of inhaled colistin as a single therapy for *P. aeuruginosa*-induced VAP, although it could be considered as combined therapy with systemic antibiotics [56,68]. The combination of antibiotics and phages is called phage–antibiotic synergy (PAS); the additive effect of both therapies increases the bactericidal effect [23]. Antibiotics and phages act in different ways, which can also lead to increasing antibiotic resistance by damaging the bacterial DNA. The main advantage of combined therapy is the use of lower antibiotic doses which are less toxic [69].

Phage therapy is a promising option to treat implant-associated infections such as VAP-related severe sepsis, especially in the presence of MDR or XDR *P. aeruginosa* strains. 

Multiple severe adverse events are caused by therapeutic high doses of antibiotics (such as nephrotoxicity which lead to dialysis, and neurotoxicity expressed by epilepsy) where clinical improvement has not been achieved. The therapeutic sequence of, firstly, antibiotics only, then a combination of immunostimulation and antibiotics was not enough to observe clinical improvement. By the addition of phages, we saw positive cutaneous graft evolution and respiratory improvement. By the second phage administration, the cutaneous graft status kept on improving and we were able to extubate the patient. Between the previous failure and the clinical improvement, the only therapeutic change was the administration of phages, so we could conclude that the association was beneficial and led to our therapeutic success.

The PhagoBurn clinical trial published in 2018 reported interesting data about phage therapy to treat *P. aeruginosa* wound infections in burn patients, but only with topical application [70]. In this trial, a cocktail of phages decreased the bacterial burden in burn wounds, but to a lesser extent compared with the standard of care. Interestingly, phage susceptibility at day 0 was significantly associated with reaching the primary endpoint by day 7. Rather than a failure of phage therapy, this trial highlights the need for personalized phage therapy with a selection of phages active on the patient’ strain based on a phagogram. Lessons learnt from this trial, as well as others, prompted us to perform a phagogram on the strains infecting all our patients. Indeed, since the conduct of this clinical trial and the development of phages in France, a large number of patients in dead-end clinical situations have been treated with phages at the referral center for the management of complex bone and joint infections (http://www.crioac-lyon.fr/en, accessed on 1 March 2024) at Hospices Civils de Lyon. This growing experience led us to launch the PHAGE*in*LYON *Clinic* program, which aims to accelerate access to phage therapy for patients experiencing severe bacterial infections (not only bone and joint infections). The previous experience of phage therapy in bone and joint infections in terms of its safety and preparation modalities facilitated its use in our burn unit under the supervision of the French National Agency for Medicines and Health Products Safety. 

To treat our burn patient, we used both inhaled and intravenous routes of administration, as the pathogen was responsible for recurrent lung infections and bacteremia. We did not treat the patient with local phage therapy on the infected burn injuries, as described in the PhagoBurn clinical trial, as we instead focused on the short-term life-threatening recurrent infections of pneumonia and bacteremia. We aimed for patient extubation and, as a secondary outcome, positive cutaneous graft evolution. Immediate phage therapy during *P. aeruginosa*-induced VAP was not feasible, as a delay is required between the identification of the pathogen and the treatment of the patient with active phages. Indeed, after microbiological identification, which usually takes a few days, it is necessary to perform a phagogram to test the activity of potential phages, as the spectrum of activity of phages targeting *P. aeruginosa* is variable. Here, by anticipating this delay, and due to the high probability of relapse in this immunosuppressed patient, a phagogram was requested for the strain responsible for the second VAP in order to identify the active phages that could be rapidly used to treat a subsequent relapse. 

Finally, it is important to note that the curative outcome of the patient was concomitant to phage therapy, antibiotics, and his immune recovery. In this severely burned patient, the extent of his immunosuppression measured by m-HLA-DR partially explained the repeated relapse of VAP due to *P. aeruginosa*. Indeed, the down-regulation of innate immunity is associated with poor prognosis in burn patients [42], whereas a competent immune system and especially functional neutrophils seem to increase phage activity. In this way, a recent report on animal models showed that neutrophil–phage synergy is essential for the resolution of pneumonia. In the case presented here, a subsequent interferon-γ injection before the third episode of *P. aeruginosa* VAP [71] was associated with a mild and transient increase in m-HLA-DR, whereas this biomarker significantly increased after the combined phage and antibiotic therapy. This initial recovery of the immune system did not prevent the occurrence of a subsequent relapse, which was the fourth episode of XDR *P. aeruginosa* pneumonia and bacteremia, although it was less severe when the patient was extubated. The patient also responded well to the treatment, showing a continuous improvement in m-HLA-DR before and after this last treatment (Figure 1).

## 4. Conclusions

The management of infections due to *P. aeruginosa* represents a major public health issue, especially in burn ICUs where the emergence of XDR isolates limits the therapeutic options. The major efforts required to fight *P. aeruginosa* should prioritize prevention, mainly through hygiene measures in health establishments to prevent the occurrence of nosocomial infections. However, ICU patients and especially severely burned patients are particularly vulnerable to *P. aeruginosa* infections. Indeed, these patients are frequently colonized by this pathogen and often experience skin graft infections and lysis, with VAP and bacteremia being facilitated by burn-induced innate immune system dysfunction. In combination with antibiotics and immune stimulation, phage therapy represents a promising option to treat these patients. In this case, the synergic effect of phage administration and lower doses of antibiotics might have been the reason for the positive outcome. Phages permitted lower doses of antibiotics and reduced their toxicities, which were serious and did not control the respiratory colonization. Although PAS did not result in microbial success, the clinical outcome was favorable. Immunostimulation may also influence the outcome. Facilitating the use of phage therapy on the basis of compassionate use is the first step before carrying out therapeutic trials in order to demonstrate its added value for this indication. This patient-adapted treatment was possible because of the low immune response to infection and the availability of the phages ready to use, with a compatible phagogram on the patient’s *P. aeruginosa* strain. 

## Figures and Tables

**Figure 1 viruses-16-01080-f001:**
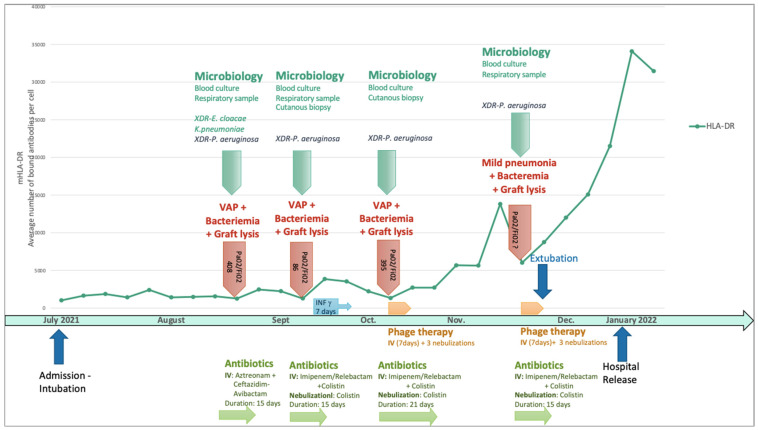
Sequential *P. aeruginosa* infections in relation to the human leucocyte antigen–DR isotype and timeline of immunostimulation, antimicrobial therapy, and phage therapy. IV: intravenous administration; VAP: ventilator-associated pneumonia; XDR: extensively drug-resistant. Pa02/Fi02: partial arterial pressure of oxygen (Pa02) over the fraction of inspired oxygen.

**Figure 2 viruses-16-01080-f002:**
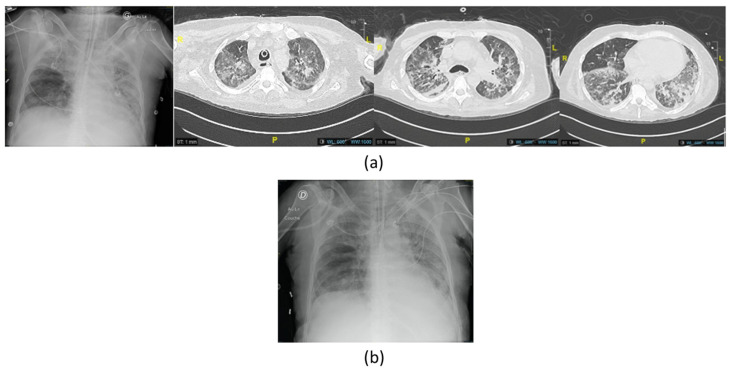
Imaging of the third episode of VAP due to XDR *P. aeruginosa* with (**a**) chest X-ray showing right apical opacities and computed tomography scan (CT scan) before the first phage therapy showing bilateral extensive diffuse pneumonia of the lungs; (**b**) chest X-ray at the end of the treatment showing improvement.

**Figure 3 viruses-16-01080-f003:**
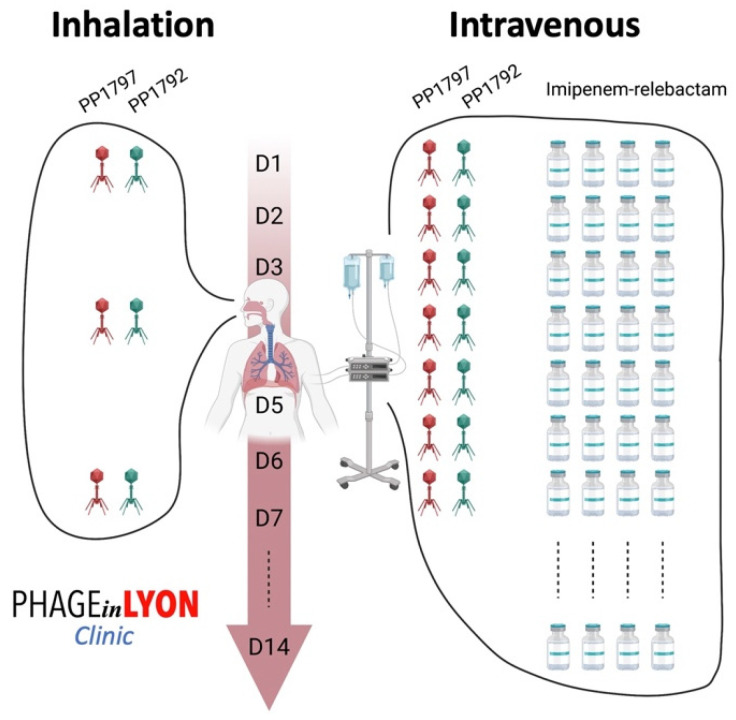
Phage administration scheme used to treat the patient, as adjuvant to imipenem-relebactam. The cocktail of active phages was administered daily intravenously and, at the same time, were also delivered locally by using a vibrating mesh nebulizer connected to the ventilator.

**Figure 4 viruses-16-01080-f004:**
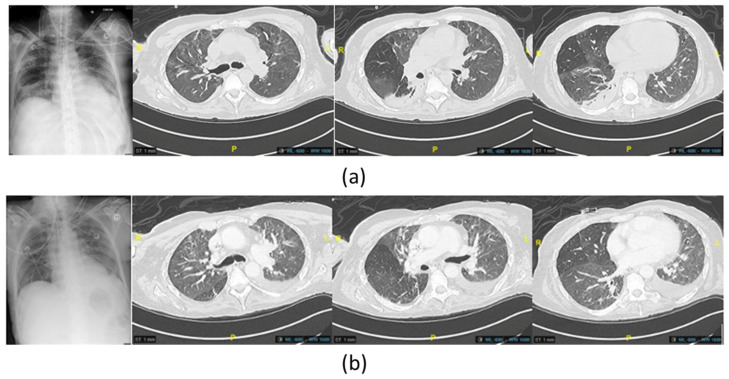
(**a**) Imaging chest X-ray and computed tomography scan of the last *P. aeruginosa* lung infection, showing new appeared posterior consolidations of the basal right lung before the administration of phage therapy (**b**) and at the end of the treatment, showing improvement in the posterior region of the right lung.

**Table 1 viruses-16-01080-t001:** Antibiogram table of each *P. aeruginosa* responsible for hospital-acquired pneumonia during the patient’s stay in the burn intensive care unit, with the minimal inhibitor concentration (MIC) of antibiotics. The term “multidrug resistant” (MDR) is determined by the non-susceptibility of a pathogen to one or more agents in three antimicrobial classes. The term “extensively drug resistant” (XDR) is determined by the non-susceptibility of a pathogen to more than one antimicrobial agent in all antimicrobial categories, except with two or less. The term “pan-drug resistant” (PDR) is determined by the non-susceptibility to all antimicrobial agents in all antimicrobial categories [29]. All *P. aeruginosa* isolates collected from the patient reported here were XDR isolates. S: Sensitive, R: resistant.

	First VAP + Bactermia + Graft lysis	Second VAP +Bactermia + Graft lysis	Third VAP + Bacteremia	Mild Pneumonia + Bacteremia+ Graflt lysis
**Ticarcilline**	R	R	R	R
**Ticarcilline + Clavulanate**	R	R	R	R
**Piperacilline**	R	R	R	R
**Piperacilline + Tazobactam**	R MIC > 32	R MIC >32	R MIC >32	R MIC >32
**Ceftazidime**	R	R	R	R
**Aztreonam**	R MIC > 32	R MIC > 32	R MIC > 32	R
**Imipenem**	R MIC > 8	R MIC > 8	R MIC > 8	R MIC >8
**Meropenem**	R MIC: 16	R MIC: 16	R	R
**Tobramycine**	R	R	R	R
**Ciprofloxacine**	R	R	R	R
**Levofloxacine**	R	R	R	R
**Cotrimoxazole**	R	R	R	R
**Colistine**	**S MIC: 2**	**S MIC: 1**	**S MIC: 2**	**S MIC: 2**
**Ceftazidim + Avibactam**	**S MIC: 4**	R MIC: 16	R MIC: 16	R MIC: 16
**Ceftolozane + Tazobactam**	R MIC: 4	R MIC: 4	**S MIC: 4**	**S MIC: 4**
**Cefepim**	R MIC > 16	R MIC > 16	R MIC > 16	R MIC: 16
**Cefiderocol**	**S MIC: 0.500**	**S MIC: 0.500**	**S MIC: 0.500**	**S MIC: 0.500**
**Imipenem + Relebactam**	**S MIC: 2**	**S MIC: 4**	**S MIC: 2**	R MIC > 8
**Meropenem + Vaborbactam**	R MIC: 16	R CMI: 16	R CMI: 16	R CMI: 16
**Tobramycine**	R MIC >4	R MIC >4	R MIC >4	R MIC > 4
**Amikacine**	R MIC > 32	R MIC > 32	R MIC > 32	R MIC > 32
**Tigecycline**	R MIC > 1	R MIC > 1	R MIC > 1	R MIC > 1
**Eravacycline**	R MIC > 0.500	R	R MIC > 0.500	R MIC > 0.500
**Fosfomycine**	**S MIC: 64**	R		

## Data Availability

Data are unavailable due to privacy and ethical restrictions.

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
