# Peer review of "Phage Therapy in a Burn Patient Colonized with Extensively Drug-Resistant Pseudomonas aeruginosa Responsible for Relapsing Ventilator-Associated Pneumonia and Bacteriemia"

_viruses, 2024, doi:10.3390/v16071080_

Round 1
Reviewer 1 Report
Comments and Suggestions for Authors
General comments:
This is a very clear and detailed clinical case report of a challenging case involving the use of phage therapy in combination with other existing treatment modalities such as intravenous and nebulised antibiotics in conjunction with immunotherapies such as interferon gamma. It describes a novel approach to managing drug resistant infection and adds to the limited literature base on this topic.
The clinical case details are extremely detailed and could be made more concise to allow for further detail on specific details regarding dosing and monitoring during phage therapy here with more focus in the discussion on limitations of the case that prevent conclusion of the true impact of phage therapy in this case. The discussion could be expanded further with more reference to the existing literature on phage use in pseudomonas infections and potential limitations here that could have contributed to initial infection relapse / recrudesence e.g. phage dosing / duration / administration routes and lack of antibiotic synergy testing and less information about pseudomonas drug resistance mechanisms and HLA-DR.
Specific comments:
1. Lines 58-59: Studies have shown this is not a reliable method for detection or diagnosis of pseudomonas infection clinically and is open to multiple biases. Would either expand on this fact or remove from the introduction
2. lines 78-82: Epidemiological resistance date references 2015-7 rates. Would suggest updating with more recent data references and figures here.
3. Lines 84-85: Can authors expand if this is infection specific or all cause mortality,
4. line 129: Would shorten this to XDR NDM-producing E. cloacae
5. Table 1: MIC value for piperacillin-taxobactam in 1st column is missing.
6. Lines 189-192: Was any antibiotic-phage synergy testing performed or considered on these strains?
7. Lines 211-214: Can authors specify what concentration of phages were used here e.g. 107 / 108 / 109 / 1010
8. Figure 2: Consider removal of arrows in diagram as not essential and distracting.
9. Lines 239-241: It is unclear why imipenem-relebactam was used for second VAP episode when table 1 demonstrates resistance of the isolate to this antibiotic. Can authors expand on reasoning for using this and not an alternative agent that was reported as suscptible in table 1.
10. Line 241: can more detail be provided on specific adverse events monitored for her (particularly those reported in other studies and case series using IV phage therapy) e.g. fever and liver function derrangment. Was specific monitoring for this conducted and detection of phage antibodies in serum? If not done can authors comment on reason for this.
11. Figure 4: Condensation should be changed to 'consolidation'
12. Line 256: Suggest refer to tetanus as Clostridium tetani
13. lines 375-376: Can authors expand more on findings of phago-burn study and study limitations. Given the negative findings from this study, what was the reason the authors still believe there is a role for phage therapy in pseudomonas colonised burns patients.
14: Discussion and conclusion: can authors comment on limitations that may prevent drawing conclusions on the specific impact of phage therapy in this case.
Reviewer 2 Report
Comments and Suggestions for Authors
The report reads as if the reader is present during the hospital conference on this patient, with an audience of clinical and laboratory colleagues, microbiologic researchers, residents, students aso. It is not only a case report but also a lesson about the complexity of burn wounds” treatments, especially of the most grave ones. The complete list of risks, the physiological abd immunological defense and healing processes on the patient’s site, but also on the pathogens’ site, the tools they use ( with focus on P. aeruginosa). It also shows how complex the (personalised) application of phages is. and how any expertise and experts are involved, including cooperative and sensitive authorities.
In this respect the importance of a cooperation of the multidisciplinary clinical team and a top phage laboratory, as well as a motivated patient, all with incredible endurance, are needed for success at the end. I have learneda lot..
I think that the choice to focus on the life threatening P. aeruginosa VAPs instead of phage trial on the wound itself is a very wise one, when compared to previous trials with phages.
I felt some analogies with the experimental stage of personalized immunotherapy or CAR-T experimental stage.
In addition to thgeneral observation and personal reflections, I have some small remarks.
- When discussing the relevance of nosocomial AMR in the introduction, and fitting into its “lecture character”, the term of ESKAPE for catching the big 6 might be mentioned.
- 113: because of the extensive burned area, it might be good to mention autografts, allografts or synthetic grafts
- 137: In fig 1: small letter for E. cloaca and K. pneumoniae
- Please check explanations of every abbreviation when used for the first time
- 197 Fig 2: would be good to also have comparative CT pictures in 2b (as is done in figure 4 a and b ( if available)
- 256: would be consequent to also use here the full name of Clostridium tetani instead of tetanus bacillus
Reviewer 3 Report
Comments and Suggestions for Authors
“Phage therapy in a burn patient colonized with extensively drug-resistant Pseudomonas aeruginosa responsible for relapsing ventilator-associated pneumonia and bacteremia”
The authors performed double phage therapy to a burn patient with P. aeruginosa lung infection in the aid of antibiotics and immune booster. The authors prepared as a case report.
Fortunately, the patient was better after the treatment with clear lung improvement and discharged from hospital.
The study was well organized with clinicians and a phage company, and the phage therapy has been started from Phagoburn clinical trial under the supervision of French National Agency.
How those phages were prepared to be fulfilled with the quality for clinical usage?
Didn’t the authors perform in vitro checkerboard assay of the phages + imipenem-cilastatin-relebactam and/or colistin to see the synergistic or antagonistic effect?
What does Pa02/Fi02 of 408, and its similar ones in the main text and figure legend?
Do the authors have PK/PD data of the phages?
What is the genetic information of the phages?
Reviewer 4 Report
Comments and Suggestions for Authors
This is a lovely attempt to report the human therapeutic application of phages to clear ventilator-associated pneumonia and bacteriemia caused by drug resistant Pseudomonas aeruginosa.
Major issues
1. Four phages PP1450, PP1777, PP1792, and PP1797 were explored, from which one was found to partially lyse the ineffective strain (Line 185, 188-189). Three phages, PP1450, PP1777, and PP1797 did not show any lysis on the strain (line 187). In contrast however, two phages PP1450 and PP1777 were considered inactive (line 190), and two phages (PP1792 and PP1797) are active and used for the therapy. It is not clear how the later phages (PP1792 and PP1797) were selected or any complementation or synergy derived from combining them together to provide a rationale for their application here. Using PP1792 is justifiable but together with PP1797 would just dilute the active phage and reduce efficacy.
2. There was very little information on previous phage therapeutic work on humans in the introduction. The information on lines 88-91 was not enough as outcomes were not stated and reasons for the observed outcome to indicate justification for the work conducted here and any potential improvements or lessons learnt.
3. In many places, reports of work were presented without a clear explanation of how they were carried out. For example:
a. It was indicated that the four phages were sourced from Pherecydes Pharma (Line 168). Could you please explain how the phages were purified or if commercialized, please say the product name? It is not clear how the phages were isolated, purified to remove toxins etc, any formulation done etc to ensure reproducibility of the work.
b. Are the phage genomes known to justify their therapeutic usability? Provide Accession number.
c. Line 172-175. Explain how spot test and EoP analyses were done or provide references. Was it a spot test or plaque assay that was conducted here? Spot test is slightly different from plaques assay, depending on what was done and the aim.
d. Line 185- how can plaque assay give you partial lysis without PFU? So, I assumed the phage was diluted and spotted on lawn of the bacterium, but lysis diminished across the dilutions without producing plaques? If yes, at what dilution was the last clearance observed? Without clear plaques formed this could indicate killing through other means eg lysis from without and not true phage infection. Hence, it is difficult to justify if this phage treatment led to clearance of the infection, more so that a cocktail of antibiotics was also used.
e. Line 195-198. Please explain how this was conducted, the time interval between the two anti-infectives (antibiotics and phage), how long was the antibiotics given before the phage, what dose? Use figure 3, after day 7?? Re-write this part please.
f. Line 206-209- please indicate if ethics were obtained for this work even if the work was conducted on compassionate grounds and consent was obtained as stated on lines 210-212. Please provide an ethics reference number for this work as human subject was used.
g. Line 246, yes, a favorable outcome was achieved but was that attributed to the phage treatment is a huge question to answer. Were phages isolated from the patient samples post treatment to indicate phage propagation? Was the infective Pseudomonas strain completely cleared, ie culture negative? If not how much reduction of colonization was observed and was the isolate still sensitive to the naïve phage after treatment?
h. Line 228- How did you know if the strain is still the same? Was the initial isolate sequenced? Did you test this for phage resistance??
Other minor issues are:
Line 31-32, 86-87- Say phage name in full, then, afterwards, use short form.
Figure 1- check scientific names.
Figure 2- what is the improvement in figure 2B, don’t seem to see any specific area you are referring to. It is difficult to compare since the figure 2A and 2B are nor showing the same area as shown by figure 4
Please add more explanation for figure 3 in a legend.
Table 1-Say what S and R meant in the table
Line 177- what was the reference strain for the EoP?
Not sure if figure 5 adds anu value to the work here.
Comments on the Quality of English LanguageNA
Round 2
Reviewer 4 Report
Comments and Suggestions for Authors
Please provide reference number for the ethics approval for this work as provided by the committee- the French National Agency for Medicines and Health Products Safety.
Author Response
The Institutional Review Board Statement has been updated: Ethical review and approval by ethic committee was waived due to the fact that the patient was treated in accordance with the Declaration of Helsinki, under the supervision of the French Health care authority, and as a consent was signed. The patient described in the present paper also consented to be included in the PHAGEinLYON Clinic Cohort study (23-5016; NCT 06185920).